# Tower Running—Participation, Performance Trends, and Sex Difference

**DOI:** 10.3390/ijerph17061902

**Published:** 2020-03-14

**Authors:** Daniel Stark, Stefania Di Gangi, Caio Victor Sousa, Pantelis Nikolaidis, Beat Knechtle

**Affiliations:** 1Department of Orthopedic Surgery and Traumatology, Kantonsspital Baden, 5404 Baden, Switzerland; danistark87@gmail.com; 2Institute of Primary Care, University Hospital Zurich, 8091 Zurich, Switzerland; Stefania.DiGangi@usz.ch; 3Bouve College of Health Sciences, Northeastern University, 360 Huntington Ave., Boston, MA 02115, USA; cvsousa89@gmail.com; 4Exercise Physiology Laboratory, 18450 Nikaia, Greece; pademil@hotmail.com; 5Medbase St. Gallen Am Vadianplatz, 9000 St. Gallen, Switzerland

**Keywords:** tower running, sex differences, age, running speed, vertical run

## Abstract

Though there are exhaustive data about participation, performance trends, and sex differences in performance in different running disciplines and races, no study has analyzed these trends in stair climbing and tower running. The aim of the present study was therefore to investigate these trends in tower running. The data, consisting of 28,203 observations from 24,007 climbers between 2014 and 2019, were analyzed. The effects of sex and age, together with the tower characteristics (i.e., stairs and floors), were examined through a multivariable statistical model with random effects on intercept, at climber’s level, accounting for repeated measurements. Men were faster than women in each age group (*p* < 0.001 for ages ≤69 years, *p* = 0.003 for ages > 69 years), and the difference in performance stayed around 0.20 km/h, with a minimum of 0.17 at the oldest age. However, women were able to outperform men in specific situations: (i) in smaller buildings (<600 stairs), for ages between 30 and 59 years and >69 years; (ii) in higher buildings (>2200 stairs), for age groups <20 years and 60–69 years; and (iii) in buildings with 1600–2200 stairs, for ages >69 years. In summary, men were faster than women in this specific running discipline; however, women were able to outperform men in very specific situations (i.e., specific age groups and specific numbers of stairs).

## 1. Introduction

Distance running is of high popularity and includes different distances, from 5 to 10 km [1], half-marathon [2,3], marathon [2,4], and up to ultra-marathon of different distances [5,6]. It is well-known that men are faster than women from 5 km to marathon [7], and in ultra-marathon running [8]. However, women were able to reduce the gap with men in ultra-marathon running, with increasing age and at longer race distances [9].

Stair climbing or tower running is a very specific running discipline, in which stair climbing has developed into the organized sport of tower running. Nowadays, tower running is a sport discipline that involves running up tall buildings, such as internal staircases of skyscrapers. However, tower running can cover any running race that involves a course that ascends a building.

To date, we have knowledge about the health benefits of stair climbing [10,11,12,13]. However, no data exist about participation and performance trends in tower running, and especially about the sex difference in this specific running discipline. Such information is valuable for athletes and coaches, to better understand and plan a race strategy, and also for race organizers, for insights regarding future events.

Therefore, the aim of the present study was to investigate participation and performance trends in tower running, with the hypothesis that men would also be faster than women in this discipline. Regarding age groups, we expected that women might close the performance gap in the older groups as already shown in long distance races [9].

## 2. Materials and Methods 

### 2.1. Ethics Approval

This study was approved by the Institutional Review Board of Kanton St. Gallen, Switzerland, with a waiver of the requirement for informed consent of the participants, as the study involved the analysis of publicly available data. 

### 2.2. Methodology 

There exists a tower running world association that presents all the results of the known races around the world on their homepage (www.towerrunning.com). In an older version of this homepage, there were only the results of the current year, and sometimes, of the preceding year. We contacted the person in charge at the association to find out whether he could provide us with older data as well. For some races, however, it was not possible to find the older results. For example, for the race at the Willis tower in Chicago, the results before 2018 were not available. For other races, such as the hustle up race in Chicago, the direct link did not work, but the results could be found by searching for the link to the race, which is also provided on the homepage of the tower running world association. Table 1 summarizes all considered events listed by the number of steps of the buildings.

From the race results, the year of the event, the completed time, the sex, and the name of both the athletes and the building were available. We further looked for the height of the building and the number of stairs and floors. Race time in m:sec was converted to running speed in km/h, using the height of the building. We removed observations from unknown climbers (where the name of the climber was not reported or not known) in order to correctly account for repeated measurements. We also considered multi-climbing.

### 2.3. Statistical Analysis

The outcome was the tower climbing speed (km/h). Descriptive statistics are presented as means (SD = standard deviations) by sex and age groups. T-tests were performed to assess the outcome difference between sex, overall and for each age groups. Two-way ANOVA tests were also performed to evaluate the multivariable effect of sex and age on the outcome. Then, to control also for repeated measurements and the other covariates, the effects of sex and age, together with the tower characteristics (i.e., stairs and floors) were examined more rigorously through a multivariable mixed effects model, with random effects (intercept) for climbers. The model was specified as follows:Tower climbing speed (Y) ~ [Fixed effects (X) = Sex*Age*BS (Stairs, df = 5) + BS (Floors, df = 5) + [random effects of intercept = runners]
where BS (Stairs, df = 5) and BS (Floors, df = 5) are 5 degrees of freedom (df) basis splines changing with the number of stairs and floors, respectively; Sex*Age*BS (Stairs, df = 5) denoted the three-way interaction term Sex–Age–number of stairs. Calendar year was not considered in the above model because it was not significant. 

Results of the regression model are presented as estimates and standard errors. Statistical significance was defined as *p* < 0.05. All statistical analyses were carried out with R, R Core Team (2016). R: A language and environment for statistical computing. R Foundation for Statistical Computing, Vienna, Austria (www.r-project.org/foundation/). The R packages ggplot2, lme4, and lmerTest were used, respectively, for data visualization and for the mixed model. The R code to reproduce the analysis is provided as Appendix A.

## 3. Results

Between 2014 and 2019, the total number of observations was 28,203 (24,007 climbers). However, the total number of observations, with non-missing sex, was 28,156 (23,960 climbers). The participation and men-to-women ratio is shown in Figure 1. We observed that we had a low number of participants and a high men-to-women ratio before 2017 (i.e., the number of men was three times the number of women in 2015). The highest number of participants was recorded in 2018. In fact, the number of women in 2018 was eight times the number of women in 2014, and the number of men in 2018 was four times the number of men in 2014. In 2019, the number of available observations decreased again. The men-to-women ratio reached a minimum in 2019 with 0.89, meaning that the number of women was higher than the number of men.

In Table 2, the mean performance by sex and age group is reported. Men were faster than women in each age group (*p* < 0.001 for all ages until 69 years, *p* = 0.003 for ages >69 years), and the difference in performance stayed around 0.20 km/h, with a minimum of 0.17 at the oldest age. In Table 3, summary statistics of performance, together with tower characteristics: height, number of floors and stairs are reported by sex. Overall, the sex difference in performance was significant (*p* < 0.001); sex differences were also significant (*p* < 0.001) in average floors and stairs climbed. The results of the multivariable statistical analysis are displayed in Figure 2, to allow an easier interpretation and understanding. Moreover, we had no significant difference between men and women alone, but in the interaction with age groups and stairs climbed (Appendix A). The variability, in terms of performance, was greater in very young and very old age groups (<20 years, 60–69 years, and >69 years). This also had an effect on sex differences. Women performed better than men in the following situations: (i) smaller buildings (<600 stairs), for ages between 30 and 59 years and >69 years; (ii) higher buildings (>2200 stairs), for age <20 years and ages between 60 and 69 years; and (iii) buildings with 1600 to 2200 stairs, for age >69 years. In all other cases, men performed better than women, with the sex difference reducing when the number of stairs increased. In Figure 3, the effect of the number of floors on performance, by sex, is shown. When the number of floors increased, the average speed of tower climbing decreased, but then increased around 90 floors, and decreased again in climbing the highest buildings.

## 4. Discussion

The aim of the present study was to investigate participation trends, performance trends, and trends in sex difference in tower running, with the hypothesis that men would be faster than women in this discipline. The main findings were: (*1*) more men than women competed before 2017, (*2*) men were faster than women in each age group and the difference in performance stayed around 0.20 km/h, with a minimum of 0.17 km/h at the oldest age, and (*3*) women aged between 30 and 59 years and >69 years performed better than men in smaller buildings (<600 stairs).

### 4.1. Change in the Men-to-Women Ratio Across Years

Before 2017, we observed a low number of participants and a high men-to-women ratio. The highest number of participants was recorded in 2018. In 2019, the number of participants decreased again and the men-to-women ratio reached the minimum of 0.89, which means that the number of women was higher than the number of men. This could also be due to a selection bias. At the time of the data collection (2017–2019), there were more results available from the earlier races and since the aim of the selection was to represent the sport and include the most important races all over the world, we did not pay attention to compare for every year the exact same number of races. This fact should encourage race directors to join the ‘Towerrunning World Association’ (www.towerrunning.com), in order to build up a firm data base for future analyses.

Generally, in races of long traditions, the men-to-women ratio is > 1.0, indicating that more men than women competed [14], but the men-to-women ratio can decrease over the years, indicating that the number of women increased over time [15]. Future studies with larger data sets are needed to investigate this trend.

### 4.2. Sex Difference in Performance

Looking at the anatomical aspect of sex difference, studies have shown that there are differences in the anatomy and physiology of the heart [16], and in the oxygen uptake in repetitive muscle activity [17] between men and women. This fact suggests that there must also be differences in performance between genders in the sport of tower running.

Men were faster than women in each age group and the difference in performance stayed around 0.20 km/h, with a minimum of 0.17 at the oldest age. However, women outperformed men in the following situations: (i) smaller buildings (<600 stairs) and ages between 30 and 59 years and >69 years; (ii) higher buildings (>2200 stairs) and ages <20 years and between 60 and 69 years; and (iii) buildings with 1600–2200 stairs and ages >69 years.

When the number of floors increased, the average running speed of tower climbing decreased, but then increased around 90 floors, and decreased again in climbing of highest buildings. A possible explanation for this fact could be the diversity of the runners. One could think that recreational runners take part in races until a certain height, because of their estimated stamina. Therefore, their running speed decreases until they reach their maximum of the height of the building. More professional runners again might only start in the races in which they have to climb the higher buildings, starting around 90 floors. Again, these professional runners will have to decrease their average running speed, to be able to climb even the highest building. This, on the other hand, is only a hypothesis that we did not investigate, and would need further studies to be verified.

Another explanation could be the men-to-women ratio by age group. When female and male age group ultra-marathoners were investigated, women could close the gap to men in older age groups (>60 years) and longer race distances (i.e., 100 miles compared with 50 miles) [9]. This relative improvement in female performance at higher ages is most likely due to the change in the men-to-women ratio in older age groups. It has been shown for female and male age group freestyle swimmers, from 25–29 to 85–89 years, competing in the FINA World Masters Championships between 1986 and 2014, that women were faster than men for age groups 80–84 and 85–89 years. When the trend for the men-to-women ratio for age groups 25–29 to 75–79 years (i.e., men were faster than women) and age groups 80–84 to 85–89 years (i.e., women were faster than men) was analyzed, the men-to-women ratio remained unchanged in 50 m, 100 m, and 400 m in age groups 25–29 to 75–79 years, but increased in 200 m and 800 m. For age groups 80–84 to 85–89 years, the men-to-women ratio remained unchanged in 50 m and 100 m, but decreased in 200 to 800 m [18]. However, in the present tower runners, the men-to-women ratio increased with increasing age, but was lowest in the youngest age group (Table 2).

Other variables could explain that women outperformed men in some specific situations (e.g., specific age groups and building heights) of this running discipline. Generally, women are lighter than men [19,20,21], which might help in running upwards. Body mass was, however, not predictive in female mountain ultra-marathoners [21]. Unfortunately, body mass was not available in these runners. Another explanation could be the motivation of female athletes [22]. For example, motivation differs between female and male marathon runners [22]. It has been shown that female marathon finishers exceeded men on the motivational scales for body weight concern, affiliation, psychological coping, life meaning, and self-esteem, and they scored lower on competitive motivation [23]. Future studies might investigate the motivation of female and male tower runners by age group and performance level.

Regarding the health aspect, it has already been investigated that stair climbing brings certain benefits. It could be shown that it helps decrease blood glucose levels [12] and that it brings a cardiac benefit in senior citizens [13]. Therefore, there is a certain interest in investigating this subject regarding public health.

## 5. Conclusions

Men are generally faster than women in tower running, but women are closing the gap with men, with increasing stairs and increasing age. The reason for the better performance in women with increasing stairs remains unclear and might be a subject for further research. 

## Figures and Tables

**Figure 1 ijerph-17-01902-f001:**
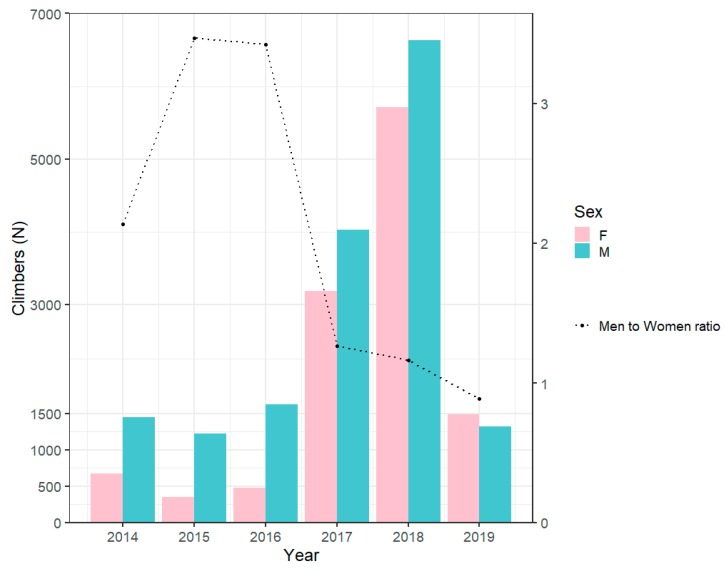
Participation and men-to-women ratio.

**Figure 2 ijerph-17-01902-f002:**
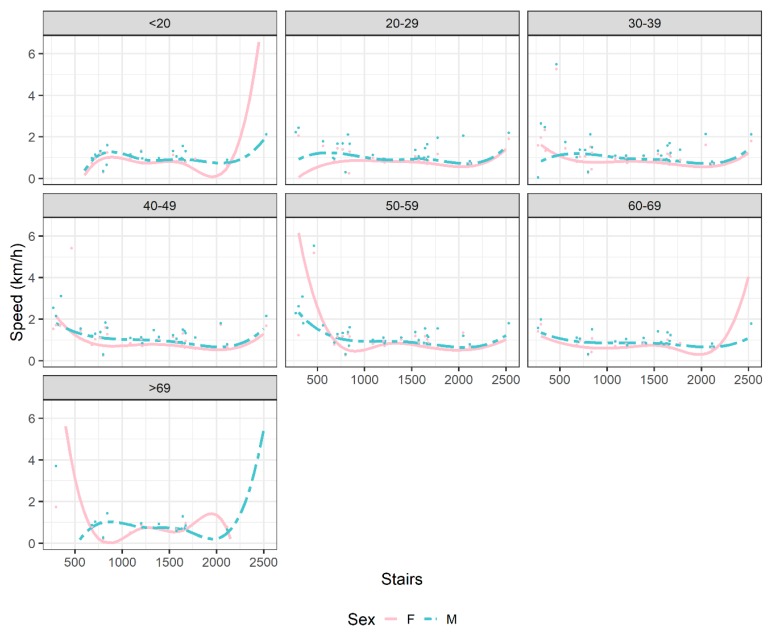
Speed (km/h) by stairs, age, and sex. Lines represent the predicted values from the mixed model and points represent the average of the observed values.

**Figure 3 ijerph-17-01902-f003:**
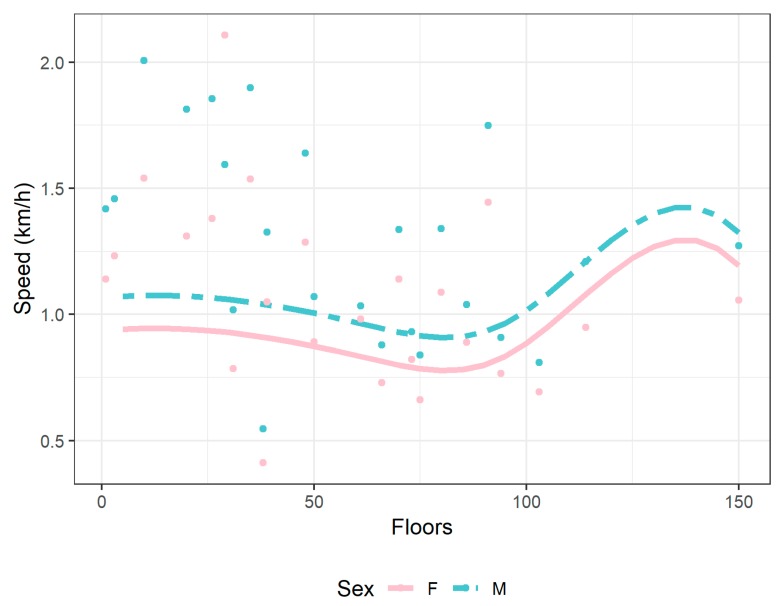
Speed (km/h) by floors and sex. Lines represent the predicted values from the mixed model and points represent the average of the observed values.

**Table 1 ijerph-17-01902-t001:** Data included in the present study.

Building	City	Steps	Data Available (Years)	Included (Years)
Millennium Tower	Wien	2529	2014–2016	2014–2016
Willis Tower (Sears Tower until 2009)	Chicago	2109	2014–2019	2018
Taipei 101	Taipeh	2046	2014–2019	2017–2018
CN Tower	Toronto	1776	2014–2019	2017–2018
Reunion Tower	Dallas	1674	2018–2019	2018
Eiffelturm	Paris	1665	2015–2020	2015–2018
AON Center	Chicago	1643	none on towerrunning.com	2018
John Hancock Center (875 North Michigan Avenue)	Chicago	1632	2014–2019	2017–2018
Empire State Building	New York	1576	2014–2019	2017–2014
Bank of America Plaza	Dallas	1540	none on towerrunning.com	2018
US Bank Tower	Los Angeles	1500	2014–2019	2018
thyssenkrupp Testturm	Rottweil	1390	2018–2019	2018
Swissôtel The Stamford	Singapur	1336	2014–2018	2017
Rockefeller Center	New York City	1214	2014–2016, 2018, 2019	2019
MesseTurm	Frankfurt am Main	1202	2014–2019	2014–2017
Three Logan Square	Philadelphia	1088	2014–2019	2014, 2018, 2019
Valliance Bank	Oklahoma City	837	2014–2019	2019
Holmenkollbakken	Oslo	800	2015–2018	2015–2017
Run Up Berlin (Park Inn Hotel)	Berlin	770	2015–2019	2015–2018
KölnTurm	Köln	714	none on towerrunning.com	2016–2019
Oakbrook Terrace Tower	Oakbrook	680	2014–2020	2019
Münsterturm	Ulm	560	none on towerrunning.com	2014–2018
Towerrun	Berlin	465	2014–2020	2018
St.George’s Tower	Leicester	351	none on towerrunning.com	2018
Matzleinsdorfer Hochhaus	Wien	342	2017	2017
Windradlauf	Lichtenegg	300	2014	2014
Haus des Meeres	Wien	271	2015–2019	2016–2018
Oluempia Hotel	Tallinn	N/A	2015–2019	2017

**Table 2 ijerph-17-01902-t002:** Summary statistics of tower climbing performance, running speed (km/h), by sex and age groups. *p*-values from t-tests for each subgroup are reported. *p*-values from ANOVA were both *p* < 0.001 for sex and age. Men-to-women ratio, computed with the number of participants, is reported.

Age Group	Sex	N	Mean (SD)	*p*	Men-to-Women Ratio
<20	F	501	0.73 (0.27)	<0.001	1.30
	M	652	0.91 (0.38)		
20–29	F	1887	0.81 (0.24)	<0.001	1.39
	M	2615	0.99 (0.35)		
30–39	F	2552	0.80 (0.30)	<0.001	1.34
	M	3415	1.03 (0.39)		
40–49	F	1941	0.78 (0.32)	<0.001	1.33
	M	2583	1.00 (0.39)		
50–59	F	1220	0.76 (0.30)	<0.001	1.60
	M	1951	0.97 (0.38)		
60–69	F	239	0.72 (0.25)	<0.001	2.62
	M	626	0.90 (0.27)		
>69	F	44	0.66 (0.33)	0.003	4.57
	M	201	0.83 (0.33)		

**Table 3 ijerph-17-01902-t003:** Summary statistics of running speed (km/h) and race time (min), tower height (m), floors, and stairs by sex. Data expressed as mean (± SD).

	Females (n = 11,886)	Males (n = 16,270)	*p*-Value
Speed km/h	0.85 (0.37)	1.06 (0.46)	<0.001
Time (min)	24.26 (14.16)	18.43 (11.69)	<0.001
Tower height (m)	296.25 (111.37)	276.27 (108.72)	<0.001
Floors	85.44 (36.37)	76.00 (35.97)	<0.001
Stairs	1466.43 (420.36)	1401.18 (429.59)	<0.001

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
