# Peer review of "Tower Running—Participation, Performance Trends, and Sex Difference"

_ijerph, 2020, doi:10.3390/ijerph17061902_

Round 1

Reviewer 1 Report

Intro

This sets the scene as to why there might be a gap in the literature on the subject of tower climbing and sex differences.

Methods and results

Table 4 does not need to be in the manuscript, consider including as supplementary. The table also needs appropriate headings.

Figure 2 and 3 the axes should be labelled bigger. The colours in figure 2 should be distinguishable in black and white. The points and line in figure 3 could be made thicker and bigger

Also the results need to be presented in a more cohesive way, and discussed in relation to what they show and then how these are interpreted, this is not so clear at the moment.

Since the analysis is conducted in R, consider providing the analysis code that is used to ensure reproducibility, since the data for this project will likely continue to expand.

Line 73: the model would be better referred to as a multivariable mixed effects model, with a random effects (intercept) for runners

Discussion

Is the conclusion based on sound knowledge or the small amount of data in the specific areas referred to where women out perform men. The authors begin to discuss some options for this, I would like some further explanation as to why the results then lead to such conclusions been drawn.

Author Response

Reviewer 1

Intro

This sets the scene as to why there might be a gap in the literature on the subject of tower climbing and sex differences.

Answer: We agree with the expert reviewer and that is the reason why we wanted to investigate this fact. We explain in the Introduction ‘To date, we have knowledge about the health benefits of stair climbing [10-13], however, no data exist about participation and performance trends in tower running and especially the sex difference in this specific running discipline. Such information is valuable for athletes and coaches to better understand and plan a race strategy, and also for race organizers for insights regarding future events’

Methods and results

Table 4 does not need to be in the manuscript, consider including as supplementary. The table also needs appropriate headings.

Answer: We thank the reviewer for the advice. We have added the headings in the Table and we have included it as supplementary Table 1, as the reviewer suggested.

Figure 2 and 3 the axes should be labelled bigger. The colors in figure 2 should be distinguishable in black and white. The points and line in figure 3 could be made thicker and bigger

Answer: We thank the reviewer for having remarked this. We have changed the Figures as suggested.

Also, the results need to be presented in a more cohesive way, and discussed in relation to what they show and then how these are interpreted, this is not so clear at the moment.

 Answer: We agree with the expert reviewer and after the adjustments we made because of other reviewers’ opinions we think the results are more comprehensible presented in the discussion.

Since the analysis is conducted in R, consider providing the analysis code that is used to ensure reproducibility, since the data for this project will likely continue to expand.

 Answer: We agree with the expert reviewer and we have provided, as supplementary, the analysis code to reproduce the main analyses.

Line 73: the model would be better referred to as a multivariable mixed effects model, with a random effects (intercept) for runners

Answer: We thank the reviewer for this suggestion. We have changed the sentence accordingly.

Discussion

Is the conclusion based on sound knowledge or the small amount of data in the specific areas referred to where women out perform men? The authors begin to discuss some options for this, I would like some further explanation as to why the results then lead to such conclusions been drawn.

Answer: Our conclusion is based on the statistical model analysis (Fig. 2-3 and supplementary Table 1) which revealed these specific situations in which women outperform men as described in the results. As other studies have shown there is a beneficial effect on health doing exercises, e.g. climbing stairs. Men normally have a higher blood volume which provides a better oxygen supply for the muscles needed to be faster in climbing the stairs. These anatomical facts are not that different in children comparing the gender and in the older ages they assimilate again. That correlates with our findings saying that in these groups men were not able to outperform women.

Reviewer 2 Report

Title manuscript:

I think that title manuscript is not correct in relation with the content of manuscript. Please, reconsider it. For example: “ Participation, performance trends and sex difference in tower running”.

Abstract:

Line 18. Please change “,” to “.” in numbers. Revise all the manuscript.

Introduction

Line 38. Change “sports” to “sport”.

Line 46. What is your hypothesis about differences between groups age?

Material and Methods:

Line 155. Could you make table 4 more visual and reduced?

Author Response

Reviewer 2

Title manuscript:

I think that title manuscript is not correct in relation with the content of manuscript. Please, reconsider it. For example: “Participation, performance trends and sex difference in tower running”.

Answer: We agree with the expert reviewer and changed the title similar to the reviewer’s recommendation. The title reads now ‘Tower running – Participation, performance trends and sex difference’.

Abstract:

Line 18. Please change “,” to “.” in numbers. Revise all the manuscript.

Answer: We thank the reviewer for this comment. Anyway, we could not change it as suggested, since the comma is the thousands separator UK/USA standard, required from the journal.

Introduction

Line 38. Change “sports” to “sport”.

Answer: We thank the reviewer for the attention to details. We have corrected it as suggested.

Line 46. What is your hypothesis about differences between groups age?

Answer: We agree with the expert reviewer and added our hypothesis about the performance in the different age groups. It reads now ‘Therefore, the aim of the present study was to investigate participation and performance trends in tower running with the hypothesis that men would also be faster than women in this discipline. Regarding age groups we expected that women might close the performance gap in the older groups as already shown in long distance races [9].

 Material and Methods:

Line 155. Could you make table 4 more visual and reduced?

Answer: We agree with the reviewer and we have improved the table appearance. Moreover, we have included it as supplementary.

Reviewer 3 Report

This study aimed to summarize data on participation and performance in tower running races among men and women. The topic is novel, and a great sample size was examined for this purpose. Findings are quite heterogeneous, but future research might yield some light on this regard. The manuscript is overall well written. I have the following comments/suggestions:

Abstract

- Line 22. Include p-value (p<0.001)?

- Lines 23-26.  I suggest organizing these findings with “i), ii), iii)” to better explain the situations in which women outperformed men.

Introduction

- Line 32. “from 10 km…”? Actually, the shortest distance is not 10km. I suggest rephrasing this sentence.

- Line 41. Some other examples of health benefits, in case they are useful (PMID: 27547414, PMID: 24033611)

Methods

Line 86. Please, include this information about R as a citation.

Results

Line 94: Is this correct? Was the number of men eight times the number of women? And in 2014, was it "four times" higher? It seems that in 2018 the ratio is approximately one, and in 2014 is around two.

Line 95: I suggest writing something like “In 2019 the number of available observations decreased again” because it is possible that the number of participants in races did not decrease, what decreased was the data obtained from these participants. In this line, why couldn´t you access data from 2019 for most races?

Line 133-134: Do you mean that "women aged between 30-59 and >69 years performed better than men in smaller buildings (<600 stairs)"? Or women outperformed men between 30-59 years and >69 years irrespectively of the size of the building? I suggest trying to make this paragraph as understandable as possible, maybe dividing the findings with bullets or similar “i), ii), iii)…”.

Line 137: “with THE sex difference”?

Line 138: Are commas really needed before and after “on performance”?

Line 139-140: What is the available sample size for this observation of a speed reduction in the highest buildings? I would probably not highlight this finding...it seems there are only three observations for buildings over 100 floors, and the trend does not strongly support a reduction. Does it?

Table 2: I would suggest including the p-value for climbing performance. Either with multiple t-tests, or ideally with a two-way ANOVA (two factors, sex and age group). Also, please, specify I the table footnote if the men to women ratio is computed with the number of participants, or with the speed (I see it is for the number of participants, but it might lead to misinterpretation).

Figure 3: Can this trend be also presented for men and women separately? The figure can include three colors (black, red and blue).

Discussion

Line 184: Do you mean that "women aged between 30-59 and >69 years performed better than men in smaller buildings (<600 stairs)"? Or women outperformed men between 30-59 years and >69 years irrespectively of the size of the building? If it is the latter option, women outperformed men at most age groups…

Line 200: It might be good, here and in other places along the text, to divide these findings with "i), ii), iii)" for the sake of clarity.

Line 204: As stated above, I wonder on how many observations is this based, and whether a reduction of speed is actually observed (based on the three available observations for buildings >100 floors, as suggested in Figure 3). It seems that speed increases with higher buildings (<100 floors), but I would maybe not emphasize the potential reduction of speed with the highest buildings if there is no strong evidence supporting it.

Line 227: Would it be better to say “in some specific situations (e.g., specific age groups and building heights) of this running discipline?”

Author Response

Reviewer 3

This study aimed to summarize data on participation and performance in tower running races among men and women. The topic is novel, and a great sample size was examined for this purpose. Findings are quite heterogeneous, but future research might yield some light on this regard. The manuscript is overall well written. I have the following comments/suggestions:

Abstract

- Line 22. Include p-value (p<0.001)?

Answer: We agree with the expert reviewer and we have added them.

- Lines 23-26.  I suggest organizing these findings with “i), ii), iii)” to better explain the situations in which women outperformed men.

Answer: We agree with the expert reviewer and we have modified the text accordingly.

Introduction

- Line 32. “from 10 km…”? Actually, the shortest distance is not 10km. I suggest rephrasing this sentence.

Answer: We agree with the expert reviewer and changed it to 5 – 10km since we mentioned this distance from another study in the following sentence.

- Line 41. Some other examples of health benefits, in case they are useful (PMID: 27547414, PMID: 24033611)

Answer: We agree with the expert reviewer and added the suggested studies to the references.

Methods

Line 86. Please, include this information about R as a citation.

Answer: We agree with the expert reviewer and add the R-code as a supplement

Results

Line 94: Is this correct? Was the number of men eight times the number of women? And in 2014, was it "four times" higher? It seems that in 2018 the ratio is approximately one, and in 2014 is around two.

Answer: We thank the expert reviewer for having spotted this. The right sentence has now been reported: “the number of women in 2018 was eight times the number of women in 2014 and the number of men in 2018 was four times the number of men in 2014.”

Line 95: I suggest writing something like “In 2019 the number of available observations decreased again” because it is possible that the number of participants in races did not decrease, what decreased was the data obtained from these participants. In this line, why couldn´t you access data from 2019 for most races?

Answer: We agree with the expert reviewer and we have edited it accordingly. We could not access data from 2019 for most races because we stopped the data gathering in the first half of 2019 and up to then we didn’t have the same amount of results as for the former years.

Line 133-134: Do you mean that "women aged between 30-59 and >69 years performed better than men in smaller buildings (<600 stairs)"? Or women outperformed men between 30-59 years and >69 years irrespectively of the size of the building? I suggest trying to make this paragraph as understandable as possible, maybe dividing the findings with bullets or similar “i), ii), iii)…”.

Answer: We agree with the expert reviewer and we have made this paragraph understandable as suggested.

Line 137: “with THE sex difference”?

Answer: We thank the reviewer for the attention to details and we have corrected it.

Line 138: Are commas really needed before and after “on performance”?

Answer: We agree with the expert reviewer and we have removed them.

Line 139-140: What is the available sample size for this observation of a speed reduction in the highest buildings? I would probably not highlight this finding...it seems there are only three observations for buildings over 100 floors, and the trend does not strongly support a reduction. Does it?

Answer: We thank the expert reviewer for having pointed this. We have clarified that, in Figure 3 (and also Figure 2) the points represented the mean of the observed values. Indeed, we have three type of buildings with floors over 100: 1) buildings with 103 floors; 2) buildings with 114 floors and 3) buildings with 150 floors. Therefore, the average speed for each type was represented as point in Figure 3. However, we have in total 5605 observations with buildings over 100 floors: 1) 103 floors: 1730 observations; 2) 114 floors: 385 observations; 3) 150 floors: 3490 observations. So, the trend supports a reduction and we have left the finding.

Table 2: I would suggest including the p-value for climbing performance. Either with multiple t-tests, or ideally with a two-way ANOVA (two factors, sex and age group). Also, please, specify I the table footnote if the men to women ratio is computed with the number of participants, or with the speed (I see it is for the number of participants, but it might lead to misinterpretation).

Answer: We agree with the expert reviewer and we have edited the table as suggested.

Figure 3: Can this trend be also presented for men and women separately? The figure can include three colors (black, red and blue).

Answer: We agree with the expert reviewer and we have presented the trend for men and women separately.

Discussion

Line 184: Do you mean that "women aged between 30-59 and >69 years performed better than men in smaller buildings (<600 stairs)"? Or women outperformed men between 30-59 years and >69 years irrespectively of the size of the building? If it is the latter option, women outperformed men at most age groups…

Answer: We agree with the expert reviewer and we have rephrased it accordingly: "women aged between 30-59 and >69 years performed better than men in smaller buildings (<600 stairs)". Here, in the discussion, we have reported as (3) the finding i) as defined below or above. 

Line 200: It might be good, here and in other places along the text, to divide these findings with "i), ii), iii)" for the sake of clarity.

Answer: We agree with the expert reviewer and we have done it accordingly.

Line 204: As stated above, I wonder on how many observations is this based, and whether a reduction of speed is actually observed (based on the three available observations for buildings >100 floors, as suggested in Figure 3). It seems that speed increases with higher buildings (<100 floors), but I would maybe not emphasize the potential reduction of speed with the highest buildings if there is no strong evidence supporting it.

Answer: We thank the reviewer for having remarked this. We have already addressed this problem above.

Line 227: Would it be better to say “in some specific situations (e.g., specific age groups and building heights) of this running discipline?”

Answer: We agree with the expert reviewer and we have changed it accordingly.

Round 2

Reviewer 1 Report

The authors have made the suggested changes to the manuscript which much improves the manuscript.